# Recent Achievements in the Heterogeneity of Mammalian and Human Retinal Pigment Epithelium: In Search of a Stem Cell

**DOI:** 10.3390/cells13030281

**Published:** 2024-02-04

**Authors:** Lyubov A. Rzhanova, Yuliya V. Markitantova, Maria A. Aleksandrova

**Affiliations:** Koltzov Institute of Developmental Biology of the Russian Academy of Sciences, 26 Vavilov Street, 119334 Moscow, Russia; 9303923@gmail.com (L.A.R.); mariaaleks@inbox.ru (M.A.A.)

**Keywords:** retinal pigment epithelium, RPE, morphometry map, heterogeneity, cell subpopulation, RPE stem cells (RPESCs), reprogramming

## Abstract

Retinal pigment epithelium (RPE) cells are important fundamentally for the development and function of the retina. In this regard, the study of the morphological and molecular properties of RPE cells, as well as their regenerative capabilities, is of particular importance for biomedicine. However, these studies are complicated by the fact that, despite the external morphological similarity of RPE cells, the RPE is a population of heterogeneous cells, the molecular genetic properties of which have begun to be revealed by sequencing methods only in recent years. This review carries out an analysis of the data from morphological and molecular genetic studies of the heterogeneity of RPE cells in mammals and humans, which reveals the individual differences in the subpopulations of RPE cells and the possible specificity of their functions. Particular attention is paid to discussing the properties of “stemness,” proliferation, and plasticity in the RPE, which may be useful for uncovering the mechanisms of retinal diseases associated with pathologies of the RPE and finding new ways of treating them.

## 1. Introduction

The retina is a specialized light-sensitive tissue in the eye of mammals and humans that provides visual perception, and is actively studied at the cellular, molecular and genetic levels [1,2]. Photoreceptor cells located in its outer part perform the function of converting light (phototransduction) into neurochemical signals, which are processed in the neurons of the retina and the brain and ultimately form our vision. Functional support for retinal neurons is provided by retinal pigment epithelium cells (RPE cells). The retinal pigment epithelium (RPE) is a single-row layer of pigmented, hexagonal, normally non-proliferating cells located between the choroid and the photoreceptor cells of the retina. The RPE performs many diverse functions to support the retina, including the transepithelial transport of substances, the phagocytosis of photoreceptor outer segments, and a number of processes in the visual cycle, as well as participation in the blood–retinal barrier and secretion of growth factors [3,4,5]. The RPE plays an important role in regulating the redox homeostasis of retinal photoreceptors [6].

Genetic disorders affecting the functioning of the RPE associated with abnormal cell proliferation and/or cell death can lead to the development of degenerative diseases of the retina, such as age-related macular degeneration (AMD), Stargardt macular degeneration (Stargardt disease, STGD), proliferative vitreoretinopathy (PVR), and others [6,7,8].

The RPE in the human eye undergoes morphofunctional restructuring throughout ontogenesis [9,10,11]. Recently, many studies have been carried out to characterize the gene profile of the RPE and retina of mammals and humans, which have made it possible to create a transcriptomic atlas of these tissues [2,12,13,14,15,16,17,18,19,20,21,22,23,24]. RPE cells demonstrate different susceptibilities to monogenic and polygenic retinal diseases, as well as to the effects of various drugs, depending on the topographic location of the cells in the RPE layer and the microenvironment [1,25,26]. It is assumed that there is a close connection between the heterogeneity of the RPE cells and the uneven distribution in the photoreceptor layer and inner layers of the retina that results from the pathological changes that occur during neurodegenerative eye diseases [1]. This review provides an analysis of the current state of the problem of RPE cell heterogeneity, based on the data from morphological studies and transcriptomics of single cells. Recent research confirms the high plasticity of RPE cells and shows that fetal RPE cells or the mature human RPE can be reprogrammed into a neuronal phenotype. For example, it is possible to reprogram fetal RPE cells into photoreceptors using small molecular factors, which opens up prospects for regenerative medicine [14,27]. In this regard, understanding the role of the entire genetic and functional diversity of cell subpopulations in the RPE, among which a special place is given to stem cells (SCs), in health and in retinal pathology determines the success of regenerative medicine.

## 2. Morphological Heterogeneity of the Retinal Pigment Epithelium

The morphological heterogeneity of eye RPE cells has been quite well studied and documented, especially recently, using modern optical and computational methods that assure high accuracy and objectivity [28,29,30,31,32]. The RPE is a monolayer of hexagonal cells lining the lower hemisphere of the eyeball, resembling a cobblestone street or honeycomb. It is known that a hexagonal shape of cells is the most stable configuration of cells of the same size. A network of hexagonal pigmented cells provides the maximum coverage of the retinal area without overlapping cells or empty areas, with the lowest surface tension [32]. In the adult human eye, the RPE monolayer consists of 4–6 million cells, which, despite their common structure, are morphologically and functionally not homogeneous. Depending on their location, RPE cells exhibit phenotypic plasticity, have great variability in the number of melanin and lipofuscin granules, and differ in the distribution of synthesized proteins. In the macula area, RPE cells are darker due to the presence of a large amount of dark brown melanin, which indicates more efficient light absorption in this area of the retina [29,33]. The human RPE contains cells with a different number of nuclei, from one to five, with an uneven distribution within the RPE cell layer. Thus, in the foveal region, where only cone photoreceptors are present, the RPE is represented by mononuclear cells, while in the perifoveal region, where rod photoreceptors predominate, multinucleated RPE cells dominate. This indicates a correlation between the multinucleation of RPE cells and the distribution and metabolism of overlying photoreceptor cells [28].

RPE cells vary in size and density. Thus, in the foveal area, the cells are small, about 12–14 µm in diameter, and their height ranges from 10 to 15 µm, while at the periphery, they have a diameter of 60 µm and are variable in height [31,34]. The density of RPE cells in the macula is higher than at the periphery [32,35,36,37]. In addition, cells differ in the structure of the apical cell membrane, namely in the shape of the villi. The shape of the villi is determined by the type of photoreceptor with which the pigment cell interacts: rod or cone. It is known that in humans, in the central area of the macula, photoreceptors are represented only by cones; they are longer and more densely packed than outside this area [38]. In this area, one RPE cell contacts a greater number of photoreceptors than outside it, so the cells are adapted to a higher level of phagocytosis of the outer segments of the photoreceptors, as evidenced by increased levels of enzymatic activity in macular RPE cells [29,39]. In addition, cellular mosaicism is observed in the RPE tissue, when neighboring cells in the same topographic region differ from each other [31]. All of the above indicates the morphological and functional heterogeneity of RPE cells.

## 3. Morphometric Studies of the Adult Human Retinal Pigment Epithelium

Morphometric studies of RPE cells have been carried out by many research groups [1,29,31,35,40,41,42,43]. The most detailed study became possible with the introduction of advanced technologies. The first complete morphometric map of the RPE of the human eye was recently created, using artificial intelligence software. The following parameters were analyzed: the cell area; the aspect ratio (AR), from one to three, where one corresponds to a cell of regular shape, whose major and minor axes are of the same length); the hexagonality (from zero to ten, where ten corresponds to a perfect hexagon); and the number of neighboring cells (from one to fourteen) [1]. From these data, heat maps were generated that allowed the identification of five RPE zones (P1–5), organized in concentric rings (Figure 1a, Table 1).

Overall, the data [1] confirmed at a new level the results of the morphometric analysis carried out by other researchers [28,31,35,40,41,42,43]. These results [1], unlike those of other authors, made it possible to see the cellular mosaic of the RPE layer in more detail and on a full scale. For example, the fovea and parafovea are visible to the eye only on heat maps, where these areas can be identified as a 3 mm wide dark blue spot (P1) in the center of a flat RPE sample, next to the optic nerve (Figure 1a) [1]. In addition, the researchers discovered a previously unknown ring of small RPE cells (P4) that were a radius of between 14 mm and 17 mm from the center of a flat RPE specimen: these cells resembled perifoveal cells in their characteristics (Figure 1a).

The morphological map of RPE cells in [1] fits perfectly with the distribution of photoreceptors in the human retina [38] (Figure 1). It was suggested that there is a relationship between RPE cell size and the ratio of rod photoreceptors to cone photoreceptors [1]. Also, the RPE cells of these topographic zones had a similar expression profile of genes associated with elastin and collagen, which was reflected in the low density of Bruch’s membrane (BM) in these areas [1,44]. In addition, they proposed that the P4 zone consists of a pool of RPE cells that retain the ability to enter the cell cycle to replenish dying RPE cells [1]. In the RPE of the peripheral part (zone P5), the largest and most elongated cells predominate (Figure 1a). These large RPE cells, with more than six polygonal neighbors, are organized in an ordered mosaic and obey the population rules for the accumulation of fluorophores, which allows them to be considered as healthy and functional cells [9]. They overlap with the peripheral ring of cones, which borders on the ora serrata [38] (Figure 1a–c). On the RPE heat map, the smaller and more densely packed RPE cells (with the bluer staining) are localized in the nasal area [1], which is reflected in a high neuronal density of 1.4–3 times that of the temporal region [45]. The topography of the multinucleated RPE cells follows that of the photoreceptors, with an almost complete absence in the fovea (where only cones are present) and a high frequency in the perifovea (with the highest density of rods). This distribution may reflect the specific metabolic requirements of the retina or other mechanisms, which are as yet unknown [28]. In general, the morphometric map provides a visual representation of the cellular heterogeneity of cells in the RPE monolayer in an adult, as well as of their possible functional load, which changes with age. For example, starting from mid-adolescence, at the extreme periphery adjacent to the ora serrata (zone P5), a strip of larger cells appears, which gradually expands and by the age of 90 occupies about 30% of the retinal area [46]. Complementing these data are studies that draw attention to the individual variability of RPE cells by showing changes in the cell area of the RPE between individuals [32]. Therefore, the diversity of cell subpopulations observed in the RPE may vary between individuals. This heterogeneity is a reflection of functional age-related characteristics of RPE cells.

## 4. Molecular and Functional Heterogeneity of the Retinal Pigment Epithelium

Gene expression profiling is an effective way to gain insight into cell function. However, bulk RNA sequencing (bulk RNA-Seq) of heterogeneous tissues can only provide an average gene expression profile for all cells in the tissue, making the sequencing results difficult to interpret. In contrast, single cell RNA sequencing (scRNA-Seq) and single nucleus RNA sequencing (snRNA-Seq) have the advantage of identifying the transcriptomic profiles of individual cells to decipher heterogeneity in complex tissues [47,48].

The transcriptomic analysis showed the general gene expression profile of fetal and adult RPEs, compared with the profiles of primary cultures, cell lines, and other human tissues [12]. Recently, an RNA-Seq dataset of the human fetal retina was analyzed [13,15], and an atlas of the single cell transcriptome of the adult human retina was created [48,49]. Using the Drop-Seq technology, a scRNA-Seq of postnatal mouse retinas was performed [19,50]. By integrating transcriptomic and single cell spatial data, the first comprehensive reference spatial atlas of single cells in the mouse retina has been created [51]. The data obtained have allowed for an interspecies comparison of human, monkey, mouse, and chicken retinas, which reveals conserved and non-conserved groups of genes [52]. The transcriptomic landscape of the human retina and RPE up to the 12th week of development [53] and the human fetal retina at the stages after 12 weeks were analyzed [54]. Several studies performed a scRNA-Seq of adult human RPE cells [2,21,22]. These results made it possible to create a genetic atlas of eye RPE cells.

### 4.1. Differences of the Human Macular and Peripheral Retinal Pigment Epithelium

One of the main functions of the RPE is its participation in the maintenance of the viability and functioning of photoreceptors. The photoreceptors of the human retina are represented by rods and three types of cones, which are unevenly distributed in the photoreceptor layer of the retina. Photoreceptors in specific topographic zones have their own functional load and their own needs, which the RPE must satisfy. For example, rods in the macular and peripheral areas of the eye differ in their gene expression profiles [48]. It was shown that some genes associated with mitochondrial electron transport are highly expressed in macular rods, compared with rods in the peripheral retina, indicating intense photo-oxidative processes and possibly the need for a high degree of protection of the macular area from oxidative stress. Photoreceptor rods have a large number of mitochondria, because they require large amounts of energy to maintain a high rate of renewal of the outer segments of photoreceptors and to support the phototransduction process [48].

Depending on the topography of the RPE cells, differences in the profiles of gene expression and protein synthesis were identified. Molecular and phenotypic differences between the macular and peripheral RPE cells have been identified by several research groups [1,2,21,25,44,55,56,57,58]. It was demonstrated that the levels of gene expression in the RPE cells of the macular and peripheral regions were very different, specifically, by 4.2 times. Among these genes, 11 genes associated with differentiation, apoptosis, cellular stress, metabolism, and regulation of the extracellular matrix (ECM) were identified, which had a significantly differential expression; for example, their expression was suppressed in the RPE cells of the macular region [25]. A group of researchers reported 438 genes differentially expressed between the macular and peripheral RPE. They showed that consistent topographic differences in the gene expression profile of the human RPE account for about 1–5% of the RPE transcriptome. It is hypothesized that this subtle difference in gene expression is enough to cause RPE cells to function differently [44]. It is known that human macular RPE cells have a higher activity of acid phosphatase and cathepsin D, compared with cells in the periphery, which suggests a differential lysosomal activity [55]. At the same time, genes reflecting the functions of the sodium–potassium pump are 40–60% more active in the peripheral areas of the RPE, which reflects regional differences in the ionic regulation of this tissue [1]. It was also shown that genes associated with elastin and collagen are highly expressed in the peripheral RPE cells, where the density of Bruch’s membrane is 3–6 times higher compared with the macula [1,44]. In addition, it was demonstrated that there were differences in the expression of genes encoding proteins that control cell proliferation. In the peripheral RPE of adult mammals, a decreased expression of mTOR, which regulates cell proliferation and cell viability via regulation expression of the p27kip1, a cell cycle inhibitor, was observed. It is supposed that this may explain the resistance of the peripheral RPE to aging processes and its higher ability for proliferation. In the central RPE, the expression of the intercellular contact genes *E-cadherin* and *F-actin* was increased [57]. Macular RPE cells express higher levels of genes involved in lipid synthesis, lipid metabolism, angiogenesis, and inflammation, while peripheral RPE cells express more transporters of monocarboxylates, α-ketoglutarate, sugar, leucine, proline, and choline and have higher expression levels of ECM-related genes [1,23,25,58].

A number of studies were performed on RPE tissue without removing the choroid. These studies also showed topographic differences in gene and protein expressions depending on the area of RPE isolation [23,59,60]. The proteomic analysis of the RPE/choroid showed 4204 unique proteins for the peripheral region, 4595 for the macular region, and 4409 for the foveal region of the RPE/choroid. The elevated expression of 66 proteins in the foveal region and 251 proteins in the macular region of the RPE/choroid, compared with the peripheral region, were identified. Gene ontology analysis classified the proteins with the highest expression in the periphery as belonging to the metabolite/energy category. The proteins with the highest expression in the macula are categorized as belonging to the following areas: the cellular response to a stimulus, the participation in cellular reproduction, homeostasis, or immune system processes [59]. Using a microarray platform to determine the differential gene expression between the macular and extramacular regions of the RPE/choroid, 21 genes were identified that were grouped into functional categories including the inflammatory response or regulation of angiogenesis, as well as ECM-related genes [60]. It was demonstrated that, when comparing the gene profiles in the nasal and macular areas of the RPE/choroid, 81 genes showed an increased expression and 39 genes showed a decreased expression in the nasal area of the RPE/choroid. When comparing the temporal and macular areas of the RPE/choroid, 70 genes had an elevated expression, and 44 genes had a reduced expression in the temporal RPE/choroid. It was noted that three genes displayed an increased expression, and eleven genes displayed a decreased expression in the nasal RPE/choroid, when compared with the temporal and nasal areas of the RPE/choroid. In addition, it was shown that RPE-specific genes were expressed at lower levels in the macular sample compared with the peripheral sample, whereas endothelial-associated genes showed a slightly higher expression in the macular sample than in the periphery [23].

Recent studies using scRNA-Seq have created gene expression atlases of human RPE cells, dividing cells into the macular and peripheral RPE zones [2,20,22]. Both cell populations expressed specific RPE marker genes, such as *PAX6* and *BEST1*; however, the latter had a reduced expression in the macular RPE. In addition, the differences were as follows: in the macular RPE, the genes *ID3* (a marker of the RPE of the macular region), *CTGF*, *FST*, *KCNMA1*, and *ADIRF* were highly expressed, while in the peripheral RPE, the genes *CRYAB* (a marker of the peripheral region of the RPE), *IGFBP5*, *RARRES2*, *CRYAB*, *FBXO32*, and *CHCHD10* were elevated [20,22] (Table 2). Furthermore, the human macular RPE was divided into two more subpopulations, according to the respective gene expression profiles: clusters M1 and M2. Gene Ontology enrichment analysis (GO) showed that the M1 cluster was enriched for genes involved in cell adhesion and cell contact, while the M2 cluster was predominantly enriched for genes involved in oxidative stress and endoplasmic reticulum stress (Table 3) [2]. In addition, a pseudotime analysis revealed that subpopulations of human RPE cells exhibited a spatiotemporal appearance in development. Thus, in the macular RPE, the M1 cells are less differentiated and transform into M2 cells during their development [2]. In the peripheral RPE, the researchers identified eight cell populations (P1–6), and in the P2 cell population, subpopulations P2-1 and P2-2 were identified by differences in gene expression and functions (Table 2).

**Table 2 cells-13-00281-t002:** Gene expression profiles of the human macular and peripheral retinal pigment epithelium (RPE).

**Human Macular RPE**		
**Marker Genes**	**Functions**	**References**
** *ID3* **	Macular RPE marker. A transcription factor that prevents epithelial–mesenchymal transition (EMT) in epithelial cells [61].	[2]
** *CTGF* **	Connective tissue growth factor. Angiogenesis, cell adhesion, and cell migration Promotes EMT.	[2,62]
** *FST* **	Follistatin neutralizes the effect of activin and other members of the TGF-β superfamily and is involved in the processes of neurogenesis.	[2]
** *KCNMA1* **	This gene encodes the alpha subunit of the calcium-activated BK channel and is involved neuronal excitability.	[2]
** *ADIRF* **	A factor involved in the development of the nervous system.	[2]
** *IRX3* **	A transcription factor involved in cell growth and adhesion to the substrate.	[2]
** *KLF2* **	A transcription factor involved in the organization of the extracellular matrix.	[2]
** *FOXP1* **	A transcription factor involved in the development of the visual system.	[2,63]
** *EGFR* **	Receptor. Autophagy, phagocytosis, and proliferation of RPE cells.	[2]
** *NR1* **	Receptor. Angiogenesis and development of the nervous system.	[2]
** *CXCL14* **	A chemokine that regulates the migration of immune cells and increases angiogenesis.	[21]
** *WFDC1* **	Protease inhibitor.	[2]
** *c-KIT* ** * **reduced** *	A cell cycle gene.	[25]
** *GSTM1* ** **reduced**	A risk factor for age-related macular degeneration.	[25]
** *ALD6* ** **reduced**	ALD6 is an important enzyme in the synthesis of retinoic acid.	[25]
**Human Peripheral RPE**		
**Marker Genes**	**Functions**	**References**
** *CRYAB* **	Crystallin Alpha B. Participates in the cellular response to stimuli and the cellular response to heat stress. Peripheral RPE marker.	[2]
** *IGFBP5* **	Participates in the regulation of migration and the proliferation of smooth muscle cells, as well as in the binding of fibronectin and insulin-like growth factor I.	[2]
** *RARRES* **	Retinoic acid receptor. Participates in cell differentiation and the negative regulation of cell proliferation.	[2]
** *FBXO32* **	F-Box protein 32 that mediates ubiquitination and subsequent proteasomal degradation of target proteins.	[2]
** *CHCHD10* **	A mitochondrial protein involved in peroxisomal lipid metabolism.	[2]
** *EGFR ligands* **	Autophagy, phagocytosis, and proliferation of RPE cells.	[2]
** *NFIB* **	Participates in the organization of the basement membrane and in the transmembrane transport of sodium ions.	[2]
** *TFPI2* **	Participates in RPE proliferation in vitro.	[22]
** *IGFBP5* **	Participates in reducing neovascularization.	[22]
** *TYRP1* **	Melanin synthesis [64]. RPE subpopulation Pr 2-1 marker.	[2]
** *RLBP1* **	Associated with the visual cycle [65]. RPE subpopulation Pr 2-1 marker.	[2]

Note: RPE—retinal pigment epithelium; Pr—peripheral RPE.

Using single-cell regulatory network inference and clustering technology (SCENIC), an analysis of the regulatory genetic networks in each cell was carried out, which showed that the *NFIB* gene is active and highly expressed in peripheral RPE cells, and its target genes are involved in the organization of the basement membrane and transmembrane transport of sodium ions [2,66]. In the macular zone, RPE cells highly express transcription factors *IRX3*, *FOXP1*, *KLF2*, *TWIST1*, and *PBX1*, as well as receptors *EGFR* and *NR* (Table 2). Their expression suggests that macular RPE cells play an important role in retinal differentiation during the development and regulation of important functions such as angiogenesis, autophagy, phagocytosis, and cell proliferation [2]. The Pr2-1 subpopulation was dominated by the following genes: *TYRP1*, involved in melanin synthesis [64], and *RLBP1*, associated with the visual cycle [65].

### 4.2. Molecular Heterogeneity of Mammalian Retinal Pigment Epithelium Cells

Numerous new data on the heterogeneity of RPE cells were obtained from laboratory mammalian specimens, in particular from the scRNA-Seq of the retina of adult male mice [26,67]. Consequently, the entire RPE was divided into six clusters, the cells of which expressed common transcripts characteristic of the RPE but had differences in the expression of genes associated with known pathways reflecting the development of the RPE. These include WNT signaling pathway genes, as well as genes of the visual cycle and the accumulation of copper and other metal ions, which contribute to melanogenesis and ultimately accumulate in melanosomes. The authors explain these results by different levels of RPE cell differentiation, which is associated with cell position and mosaicism in the RPE monolayer [67].

Not only functional intercellular heterogeneity was shown, but also a heterogeneous distribution of cells in the mouse RPE layer [26]. Five cell clusters were identified in the native mouse RPE, which differed from each other in their functional specialization. Cluster 1 mouse RPE cells were enriched for genes of metabolic processes, including ATP synthesis, most likely due to the high phagocytic load of these cells. In the RPE cells of cluster 2, genes regulating processes associated with cell motility and differentiation, apparently involved in the response to environmental changes, predominated. In the RPE cells of cluster 3, processes associated with mRNA processing and splicing and melanin biosynthesis predominated, while in the RPE cells of cluster 4, catalytic processes, as well as peptidase and hydrolase activity, were most prominent. Cluster 5 is the smallest cluster of mice RPE cells, and cell functionality in this cluster has been linked with muscle-related processes that are most likely associated with the epithelial–mesenchymal transition (EMT) and cyclic guanosine monophosphate- (cGMP-) mediated signaling [26].

Functional similarities have been found between human RPE cell subpopulations and mouse RPE clusters (Table 3). In addition, similarities in gene expression changes during the development of human and mouse RPE were revealed. Specific genes showing an increased expression in the early developmental stages of human fetal RPE and mouse RPE cluster C1 (mouse RPE stem cells) include the following: *DCT/Dct*, *PAX6/Pax6*, *ID3/Id3*, and *MDK/Mdk*; the genes with a decreased expression include the following: *TTR/Ttr*, *RPE65/RPE65*, and *LRAT/Lrat* [53,67]. Based on the analysis of *Id3* expression in mouse RPE stem cells (C1) and developing human RPE cells, a hypothesis has been proposed about the existence of stem cells in the macular region of the adult human eye [67].

**Table 3 cells-13-00281-t003:** Functional similarity of human retinal pigment epithelium (RPE) subpopulations and mouse RPE clusters, based on GO data [2,26,67].

Human RPE Subpopulations [2]	GO	Description	Mouse RPE Clusters [67]	Mouse RPE [26]
**M 1**	GO:0034109	homotypic cell–cell adhesion	C1	
	GO:0031589	cell–substrate adhesion	C1	
	GO:0030198	extracellular matrix organization	C1, C2	C2
	GO:0045229	external encapsulating structure organization	C1, C2	
	GO:0034446	substrate adhesion-dependent cell spreading	C1	
	GO:0010810	regulation of cell–substrate adhesion	C1	C2
	GO:0034329	cell junction assembly	C5	
	GO:0150115	cell–substrate junction organization	C5	
**M 2**	GO:0006457	protein folding	C4	
	GO:0050821	protein stabilization	C4	
**Pr 1**	GO:0001667	ameboidal-type cell migration	C1, C5, C3	
	GO:0010038	response to metal ion	C1, C5	
	GO:0050673	epithelial cell proliferation	C1, C5	
	GO:0050678;GO:0050673	regulation of epithelial cell proliferation	C1, C5	C2
	GO:0071294	cellular response to zinc ion	C1, C5, C2	C1
	GO:2001026	regulation of endothelial cell chemotaxis	C5	
	GO:0035767	endothelial cell chemotaxis	C5	
**Pr 2-1**	GO:2001057	reactive nitrogen species metabolic process	C1	
	GO:0098869	cellular oxidant detoxification	C2	
	GO:0007601	visual perception	C4, C6	C1
	GO:0001523	retinoid metabolic process	C6	C1
**Pr 2-2**	GO:0030301	cholesterol transport	C3, C6	
**Pr 3**	GO:0042542	response to hydrogen peroxide	C5	
	GO:0030336	negative regulation of cell migration	C1	C2
	GO:0001952	regulation of cell–matrix adhesion	C5	
**Pr 4**	GO:0062012	regulation of small molecule metabolic process	C5	
	GO:0071248	cellular response to metal ion	C5	
**Pr 5**	GO:0031667	response to nutrient levels	C3, C5	
	GO:0043270	positive regulation of ion transport		C1
**Pr 6**	GO:0008203	cholesterol metabolic process	C3	
**Pr 7**	GO:0006575	cellular modified amino acid metabolic process	C1	

Note: RPE—retinal pigment epithelium; GO—Gene Ontology; M—macular RPE; Pr—peripheral RPE; and C—mouse RPE cluster. The M1 subpopulation of human RPE [2] corresponds to C1 [67] and to cluster 2 [26]; the M2 subpopulation of human RPE [2] corresponds to the C4 cluster of mouse RPE [67]; the Pr1 subpopulation of human RPE [2] corresponds to the C5 cluster of mouse RPE [67] and to clusters 1 and 2 [26]; the Pr2-1 subpopulation of human RPE [2] corresponds to the C4 cluster of mouse RPE [67] and to cluster 3 [26].

The results of studies obtained from the RPE and retina of mice cannot be directly transferred to humans. In the central part of the retina in mice, in contrast to humans and other primates, there is no clearly defined macular region that is enriched with cone photoreceptors and that has increased visual acuity, and there is no peripheral ring of cones [54,68,69]. Mice do not have a macular RPE cell subpopulation, as we understand it in humans. In addition, mouse cone cells differ from human cells in their wavelength-sensitive opsin expression patterns [48]. Despite certain similarities, there are species-specific differences in the RPE cells in gene expression patterns and in regulatory systems between humans and other vertebrates [6]. The accumulating evidence shows that there are consistent differences in the expression of a large number of genes in human and mammalian RPEs, associated with cell topography, function, and response to cellular stress [1,25,26,70].

## 5. Heterogeneity of Retinal Pigment Epithelium Derived from Induced Pluripotent Stem Cells

The hereditary nature of RPE cell specialization was confirmed by in vitro experiments [71]. Against the backdrop of actively developing studies of RPE cells derived from embryonic stem cells and induced pluripotent stem cells (iPSCs) (iPSC-RPE), the question arises about the emergence of heterogeneity similar to the native RPE in iPSC-RPE. Human pluripotent stem cells (hPSCs) have the ability to multiply indefinitely in vitro and give rise to any type of cell in the body, including the cells that form the retina. Various protocols for the differentiation of hPSCs into RPE cells have been described [72,73,74,75,76,77]. The iPSC-RPE cells are typically analyzed after several weeks of differentiation, at which stage they show similarities to the native RPE in terms of the morphology and expression of key proteins, as well as gene expression profiles. However, some characteristics of the iPSC-RPE are similar to the embryonic RPE [76,78,79].

The morphometry of iPSC-RPE cells was compared with the native adult RPE [1]. The researchers showed that the cell area of the iPSC-RPE was similar to the RPE subpopulation of the topographic zone P1, while the AR and hexagonality were similar to the P5 subpopulation. The number of neighboring cells in the iPSC-RPE was comparable to all RPE subpopulations. The authors attributed the high variability of all parameters in the iPSC-RPE group to differences in the sources of most cell lines and to diverse differentiation protocols used in different laboratories [1]. Another study showed the genetic architecture of iPSC-RPE cells under short- and long-term cultivation, demonstrating a gradual process of RPE differentiation and maturation, as well as a stable RPE phenotype during long-term cultivation [80]. In addition, it was shown that the more complex the functions of a subpopulation of RPE cells are, the later they differentiate in vivo [2,53]. It was also suggested that long-term in vitro studies of iPSC-RPE cells would allow the modeling of specific phenotypes observed in native mature RPE tissues [80].

The transcriptomic and proteomic analyses of iPSC-derived RPE cells identified a number of RPE subpopulations at different levels of maturity. Cell classification and trajectory analysis clearly indicated the efficiency of iPSC-RPE differentiation [81]. During the development from the progenitor cell stage, the iPSC-RPE cells diverged in their differentiation along two trajectories. The gene expression profiles coincided in 99.4% of the genes and differed only in the expression of 31 genes. The first trajectory differed in genes involved in the extracellular matrix (ECM) organization (*TSPAN8*), melanogenesis (*TPH1*), or retinal development (*IRX6*); and genes expressed in trajectory 2 were associated with the lipid metabolism (*ADIRF*, *APOA1*, *CD36*), iron binding (*LCN2*, *MT1G*), cytoskeleton (*MYL7*), or retinal development (*PITX3*). This divergence in iPSC-RPE cell differentiation is reminiscent of the differentiation of RPE cells from the macular and peripheral regions of the human eye in vivo [2,53].

Thus, the iPSC-RPE is a heterogeneous population of RPE cells at different stages of differentiation, which will make it possible to obtain the various functional specializations, by analogy with the native RPE.

## 6. Proliferation and Plasticity of Retinal Pigment Epithelium Cells

### 6.1. Proliferation of Retinal Pigment Epithelium Cells

Once the RPE cell differentiation is complete [82,83], the cells exit the cell cycle and persist throughout life, dividing rarely or not at all. Mammalian and human RPE cells are thought to be terminally differentiated, postmitotic, nondividing cells. However, there is evidence that in rodents, the RPE cells at the periphery of the layer are capable of proliferation [57,84]. It is also known that the peripheral RPE cells of macaques can incorporate 3H-thymidine, indicating their proliferation [28,85]. This information is complemented by data on the differential expression of genes associated with the cell cycle. Thus, it was shown that genes stimulating proliferation dominated in peripheral RPE cells [57].

In addition to exhibiting proliferative activity in situ, adult mammalian and human RPE cells can be activated to divide when placed into a cell culture, where their proliferation is facilitated by the loss of intercellular contacts and various growth factors in the culture medium. It is assumed that the proliferation and amplification of RPE cells in vitro occurs mainly due to a subpopulation of stem cells (SCs) in the RPE [86].

### 6.2. Expression of Stem Cell Markers and Differentiation Potential of the RPE Cells

The expression of multipotency markers and stem cells in the RPE is an important criterion that allows us to classify some RPE cells as adult SCs. However, in order for cells to be classified as SCs, they must possess a number of characteristics, such as self-renewal and the generation of specialized differentiated progeny. Self-renewal is the ability of SCs to proliferate symmetrically or asymmetrically and to produce a similar cell [87]. The replacement of damaged or lost cells during regeneration occurs through various mechanisms, including dedifferentiation, transdifferentiation, or reprogramming [88,89,90]. All three mechanisms can be observed in the RPE layer of the vertebrate eye. Classic experiments on model animals have shown the ability of RPE cells to transdifferentiate/undergo reprogramming into neural cells of the retina, which is successfully reproduced in vivo in lower vertebrates. Dedifferentiation is the transition of terminally differentiated cells to a less differentiated state in which cells can divide and, within their cell lineage, locally replace lost cells, moving to their final differentiated state in some tailed amphibia [89]. During the transdifferentiation of RPE cells after retinal damage in these animals, the already-differentiated RPE cells change lineage and reprogram into neurogenic progenitors expressing Klf4, Sox2, Pax6, and c-Myc [89]. The descendants of these progenitor cells differentiate into all retinal cells, including photoreceptors, glia, and pigment epithelium, and then fully restore retinal function in amphibia [89,91,92,93]. Damage to the retina in mammals and humans leads to the development of pathologies accompanied by the acquisition of a mesenchymal phenotype by RPE cells [89,93]. The loss of intercellular contacts triggers the mechanisms of EMT. After the accumulation of the necessary pool of progenitor cells, the EMT and differentiation into different cell types begin. Some researchers draw a parallel between these processes in the retina and embryonic development, as well as the development of tumors [88]. In this regard, in adult mammals and humans, RPE transdifferentiation as a means of retinal regeneration is ineffective. However, some RPE cells in mammals may exhibit proliferation, plasticity, and conversion to neurons, detectable in vitro [28,57,84,85]. During reprogramming in vitro, a fully differentiated cell can partially return to its pluripotent stage, acquiring only some characteristics of SCs, then proliferate and differentiate into other cell types. In birds and mammals, RPE transdifferentiation into retinal neurons occurs during early embryonic development, only under the influence of basic fibroblast growth factor (bFGF) or lin-28 [94].

### 6.3. Differentiation of Stem/Progenitor Cells from the Retinal Pigment Epithelium into Muscle and Adipo-, Osteo-, and Chondrogenic Cells

In a number of studies, a few cells have been isolated from the human RPE, which, according to strict clonal analysis and other stem cell criteria (self-renewal and the production of differential progeny), were classified as adult RPE stem cells: RPESCs-derived RPE (RPESCs-RPE) [95,96,97,98,99,100]. The term RPESCs-RPE was first proposed after the ability of RPESCs-RPE for multipotent differentiation was shown in the neural and mesenchymal (osteogenic, adipogenic, and chondrogenic) directions [100].

It was demonstrated that an adult RPE in vitro can generate cells expressing markers of cells of mesenchymal origin: muscle and adipo-, osteo-, and chondrogenic cells [100]. To exclude contamination and to prove the multipotency of the stem cells, the authors obtained RPESCs from a primary human RPE by cloning single cells. The cells were placed into different culture wells and exposed to adipocyte, chondrocyte, and bone differentiation media. This study was conducted on cells from a variety of sources, including human fetal RPE, RPE from various mammalian species, ARPE-19 cells, and RPE cells derived from induced pluripotent cells and SCs. The results of this study showed that mesenchymal differentiation is characteristic of the RPE. It was found that human fetal RPE is most resistant to acquiring mesenchymal fates. In recent research, stem cells from the fetal RPE (fRPESCs) were studied, obtained under the influence of vitamin C and valproic acid [27]. Cells of the fRPE and fRPESCs were placed into adipogenic, chondrogenic, and osteogenic differentiation media to test their mesenchymal differentiation capabilities. Staining cells with specific markers and quantitative real-time PCR showed that fRPESCs are more capable of differentiating into adipocytes, chondrocytes, and in the osteogenic direction than fRPE cells. The regulatory role of *SOX2* in the cellular conversion of fRPESCs was identified and it was shown that fRPESCs lose the ability for mesenchymal differentiation after the knockdown of *SOX2* [27]. Along with the differentiation of RPESCs in the mesenchymal direction in vitro, it is well known from many years of pathomorphological observations that the cartilaginous and bone formations that develop from RPE cells are sometimes found in human eyes [100,101,102]. These data indicate that RPESCs have a broad repertoire of differentiations, reminiscent of that of neural crest cells [103]. It can be speculated that this multipotency of RPESCs may be somehow related to the very early release of neuroepithelial RPE progenitor cells during embryogenesis.

There is no information about the differentiation of RPESCs in the neural direction in mammalian and human eyes in vivo, while development in the mesenchymal direction, with the formation of epiretinal membranes in retinal pathologies, is a well-known medical fact [100,104].

### 6.4. Differentiation of Stem/Progenitor Cells from the Retinal Pigment Epithelium along the Neuronal Pathway

More than 20 years ago, our laboratory established that in the total population of RPE cells of the adult human eye, the pluripotency genes *OCT4*, *NANOG*, and *SOX2* are expressed, as well as the neural differentiation gene *PAX6*, which was identified in the RPE during embryonic development, indicating that a certain proportion of cells with stem/progenitor properties are possibly present in the RPE [105,106,107]. Subsequent studies of the behavior of adult and fetal RPE cells in vitro showed that cells in culture dedifferentiate and express stem cell and poorly differentiated cell markers *OCT4*, *NANOG*, *SOX2*, *PAX6*, and *PROX1*. Under in vitro differentiation conditions, the cells express neural and glial cell markers such as NESTIN, *MSI1*, βIII-Tubulin, neurofilaments (68–200 kDa), and GFAP. In addition, using immunostaining, markers of mature retinal and brain cells were detected: recoverin (photoreceptor marker), dopaminergic neuron markers, tyrosine hydroxylase (TH) (retinal amacrine cell marker), GABAergic interneuron markers, nNOS (retinal amacrine or ganglion cell marker), as well as CNPase and O4 (oligodendrocyte markers).

Furthermore, naturally, we also detected markers of redifferentiation to the original phenotype (*RPE65, PEDF, CRALB*) [105,106,108,109,110]. Based on these data, we hypothesized that the human RPE may contain highly plastic cells that have (or acquire during the process of dedifferentiation) the SC phenotype with a dual potential. These cells may retain the potential to redifferentiate into RPE and to differentiate into retinal cells, under the influence of appropriate microenvironmental factors.

The native human RPE and cell culture have been characterized by the expression of stem cell markers, including *NANOG*, *OCT4*, *SOX2*, *SSEA-4*, *KLF4*, *C-MYC*, and *LIN-28*. Cultured RPE cells were positive for the surface marker SSEA4, showed little reaction to SOX2 immunostaining, and showed no immunostaining for OCT4 and NANOG [100]. It was found that the combined effect of vitamin C and valproic acid activates the expression of the retinal progenitor markers MITF, OTX2, and PAX6, as well as the mesenchymal stromal markers CD133, CD73, CD105, and CD90 in fRPE cells. As a result, the cells enter a retinal stem cell-like state (fRPESCs). Researchers believe that a high expression of SOX2 in fRPESCs is a prerequisite for maintaining retinal stem cell properties and a multipotent differentiation potential [27].

The search for and study of the stem properties of the RPE continues, using modern molecular genetic technologies. Among the identified mouse RPE clusters, researchers paid special attention to cluster C1, containing only 1–2% of the total number of cells [67], and cluster 5, in which only 19 cells (0.59%) were found [26], which were cells with the characteristics of stem cells and/or progenitor cells. In these clusters, a high expression of stemness and stem cell maintenance genes was observed, against the background of a high expression of melanogenesis genes. This indicated that the cells were still in the process of differentiation. It was also found that there was a high correlation of gene expression between RPE cells of the C1 cluster and retinal cells, which possibly pointed to the ability of C1 cells to differentiate into different cell types of the neural retina [67]. In addition, it was demonstrated that the RPE cells in the adult mouse eye are epigenetically very similar to the phenotypes of retinal progenitor cells and photoreceptors [111].

Mouse RPESCs were induced in vitro through the stage of sphere formation (sphere-induced RPE stem cells, iRPESCs) and it was discovered that the gene expression profile of iRPESCs was very different from the parental RPE. At the same time, changes in gene expression occurred immediately after the formation of spheres. Mouse iRPESCs expressed increased levels of *c-Myc*, *Oct4*, *c-Kit*, and *Cd44* [96]. The expression of eight of the fifteen selected stemness genes, such as *Klf4*, *Alpl*, *Kit*, *Kitl*, and *Bmi1*, was significantly upregulated in RPE spheres, and the expression of *Abcg2*, *Bmi1*, *Cd44*, *Kitl*, and *c-Myc* was upregulated in iRPESCs, compared with the original RPE. Two DNA methylation genes (*Dnmt1* and *Dnmt3a*), four histone acetylation genes (*Hat1*, *P300*, *Myst2*, and *Myst3*), and seven deacetylation genes (*Sirt2*, *Sirt6*, *Hdac1*, *Hdac2*, *Hdac3*, *Hdac5*, and *Hdac6*) were highly expressed in the stem cells, reflecting epigenetic regulation that may promote the expression of major stemness genes, such as *Oct4* and *Klf4* in iRPESCs [96,112].

The proliferation of human RPESCs was stimulated by cultivating them in a free-floating state with the addition of growth factors, similar to neural stem cells (NSCs) [100,105,106,110]. The mouse RPE cells were cultivated in the form of spheres to obtain induced iRPESCs [96]. The RPE cells were cloned in adherent cultures to expand the number of RPESCs [98], using protocols for isolating and culturing RPESCs from previous studies [95,97,99,113,114]. The human fetal RPE was treated with vitamin C and valproic acid; the stemness properties of the resulting fetal RPESCs (fRPESCs) were studied, and a significant increase in the stem cell markers SOX2, OCT4, and KLF4 was found [27]. In another study, RPESCs were activated by the influence of amniotic fluid factors, and the retinal progenitor cells obtained in this way were studied [115]. From the above studies, it follows that in the adult human and mouse RPE, cells are preserved that have signs of SCs that can be identified in vitro.

Clonal analysis of adult RPE cells was used and it was shown that primary spheres from free-floating cells were formed after 4 days with a frequency of 1.5% of the number of initial seeded cells [100]. In the adherent cultures, about 10.6% of the cells actively proliferated, although the vast majority did not divide or produced only limited progeny [100]. It was found that 10% of RPE cells proliferate once in culture, using the clonal plating. Among RPE cells, only 2% of cells proliferate very actively and can create up to 90% of the entire monolayer. These cells were classified as RPESCs [98]. If mouse RPE cells are cultured through the sphere formation stage, approximately 0.003–0.013% of RPESCs can be activated. These cells actively proliferated for more than ten passages, in contrast to other RPE cells, which showed limited proliferation and senescence after only five passages [27].

In studies of the mechanism of entry of human RPE cells into the phase of DNA synthesis in vitro, it was found that mitogen-activated protein kinase (MAPK) and extracellular signal-regulated kinase (ERK) play a key role. In this signaling pathway, growth factor receptor activates Ras GTPase, leading to MAPK/ERK phosphorylation. The MAPK/ERK signaling pathway, in turn, regulates transcription factors, c-myc, Pax6, klf4, and MITF, the expression of which indicates a decrease in the level of RPE differentiation [86,116]. Several growth factors, such as Vegf, Tgf, Pdgf, Egf, and Ngf; their receptors, Vegfr, Egfr, Pdgfr, and Tlk4; and two stem cell-associated signaling ligands, Kit-l and Lif, are involved in RPE reprogramming into induced iRPESCs and are highly expressed in iRPESCs. The ability to proliferate has been shown to be maintained through the persistent repression of cyclin-dependent kinase inhibitor (CDKI) genes, including p16, Arf, p21, and p57, which were expressed at very low levels in the RPESCs, compared with their parental RPE cells [96].

From the above studies, it follows that among the heterogeneous RPE cells of mice and humans, 0.003–2% of cells can be distinguished that are capable of proliferation, self-maintenance, and the expression of specific genes characterizing the state of stemness in cells. These cells may remain quiescent or exhibit multipotent differentiation into a mature cell type. Under expansion culture conditions in which RPESCs self-renew, they generate SC progeny, and, conversely, under differentiation culture conditions, RPESCs generate progeny of different cell types. RPESCs can produce RPE cells, and epithelial monolayers of RPE cells are easily obtained. In in vitro differentiated media that activate neural or mesenchymal differentiation, RPESCs can generate both neuronal and mesenchymal cells [86,100].

### 6.5. Differentiation of Stem/Progenitor Cells from the Retinal Pigment Epithelium into Retinal Photoreceptor Cells

In our laboratory, using qPCR analysis and immunostaining, it was shown that under in vitro differentiation conditions, the RPE cells of embryonic and adult human eyes express markers of progenitor and neural cells [105,106,108,109,110]. Other researchers have found photoreceptor markers in RPE cell lines and primary RPE cultures, such as Crx10 [117], Opn3, Opn4, Nrl, Crx, Opn1mw/1w, Sag, Nr2e3, and recoverin [118]. The identification of the recoverin protein suggests that human RPE cells are capable of differentiating into retinal photoreceptor-like cells. In that study, the human RPE cells were cultured in a medium that promotes the differentiation of retinal neurons [100]. Using qPCR analysis, it was shown that the expression of the neural progenitor marker *NES* and the neuronal marker *TUBB3* was significantly activated in RPE cells (over 1000-fold and 90-fold, respectively). Interestingly, under these conditions, the cells increased the expression of the eye field marker genes *LHX2*, *OTX2*, and *RX*, which are characteristic of the early stages of eye development, while the levels of retinal progenitor markers *CHX10* and *RHO* did not change, and the expression of *PAX6* actually decreased. These culture conditions have been noted to promote the differentiation of RPESCs toward neural progenitors of the forebrain and retina [100].

The directed differentiation of human fRPESCs into photoreceptors using a special three-step protocol using different culture systems and media has been carried out [27]. During the process of differentiation, the fRPESCs changed their morphology, first forming a round shape and then extending several synapse-like structures to finally form a tubular rod-shaped structure resembling the outer segment of a photoreceptor. At the first stage of differentiation, retinal photoreceptor progenitor cells with the expression of markers *PAX6* and *VSX2* were obtained. At the second stage, the cells were differentiated into photoreceptor progenitors that significantly increased the expression of photoreceptor markers *NRL* and *CRX*. Finally, at the terminal stage of RPESC differentiation, rod photoreceptor cells expressing *REC*, *RHO*, *ARRESTIN*, and *GNAT1* were obtained [27]. Another study managed to differentiate fetal RPE cells into rod photoreceptors by chemical reprogramming [14]. Gene ontology analysis of RNA sequencing results from these cells revealed the upregulation of genes involved in neuronal generation, neurotransmitter uptake, and photoreceptor cell differentiation, such as *SOX8*, *IGFN1*, *ASCL1*, *RXRG*, *THRB*, and *RORB*. On day 10 of reprogramming, the transcriptome profile showed a stable activation of genes specific to rod photoreceptors, but not to cones [14].

### 6.6. Differentiation of Stem/Progenitor Cells from the Retinal Pigment Epithelium along the RPE Pathway (Redifferentiation)

Mouse and human RPE stem cells are capable of redifferentiating to the original phenotype. The sphere-derived dedifferentiated RPESC-like cells can differentiate back into RPE cells in vitro [27,116]. Monolayer cultures of RPE cells derived from human RPESCs have been described and characterized as cultures with the morphology and physiology characteristic of the native RPE [95,97,99,113,114]. The redifferentiation of RPESCs occurs within 8 weeks of cultivation. During this time, the cells change their morphology, proliferation rate, and polarization and also acquire the key phenotypic characteristics of the RPE. In cultured spindle-shaped RPESCs, a significant decrease in proliferation was observed after 2 weeks; after 4 weeks, the appearance of islets of cells with a cuboidal morphology was noted, and, by week 8, almost all cells acquired the morphology of a mature RPE. The secretion of vascular endothelial growth factor A (VEGF-A) through the apical and basement membranes increased from week 2 to week 4, while the secretion of pigment epithelium-derived factor (PEDF) appeared during week 6 of the cell differentiation [95,97,99]. In the cultured adult RPE, a similar but sometimes increased expression of RPE markers was observed, compared with the native tissue. Claudin-19 was present along the apical–lateral membrane along with the tight junction protein ZO-1, indicating the existence of a functional epithelial barrier. Ezrin, a membrane-associated protein involved in cytoskeletal organization, was found predominantly in the RPE microvilli. The visual cycle proteins, cellular retinaldehyde-binding protein (CRALBP) and RPE65, were localized, as expected, in the cytoplasm. Monocarboxylate transporter 1 (MCT1) was present on the apical surface of cells [98]. This indicates the establishment of the polarity characteristic of mature RPE cells. It is obvious that, in mouse and human RPEs, cells with stem properties are present, which also have the ability to redifferentiate to the original phenotype of RPE cells.

Throughout the entire period of differentiation, the RPESC-RPE cells continuously expressed pigment epithelium cell fate determination factors OTX2 and MITF and did not express smooth muscle actin, an indicator of epithelial–mesenchymal transition. It has been shown that, during the differentiation period, transepithelial resistance increases and reaches the norm [95,99]. However, the phagocytic activity of RPESC-RPE cells decreased after 4 weeks of differentiation, apparently due to a lack of contact with photoreceptors, which are necessary for the full maturation of the RPE [97].

RNA-Seq and scRNA-Seq on differentiated RPE cells derived in vitro from human RPESCs identified 13 cell subpopulations with different functional specializations. The transcriptomic analysis showed that the subpopulation composition of the entire RPE cell population is subject to dynamics over time. These findings suggest that RPE subpopulations have overlapping but distinct functional profiles. Among these subpopulations, there are ten profiles in number. In these cells, the specific marker EZH2 (a transcription factor involved in histone methylation, self-renewal, and the differentiation of SCs) was isolated, indicating a state similar to stem cells or progenitor cells [71]. This study has confirmed RPE heterogeneity under culture conditions, similar to the native tissue, and answered the previously raised question about the innate differences between RPE cells, and whether a trait can be accurately passed on to daughter cells in the reproducing population [31]. It is likely that the mosaicism of RPE cells is hereditary in nature, embedded in the genetic program of the cells themselves, and conditioned by epigenetic influence.

### 6.7. The Potential of Retinal Pigment Epithelium Cell Subpopulations for Transplantation

Retinal degeneration as a result of the death of photoreceptors and RPE is the main cause of many degenerative diseases of the human eye, leading to vision loss [119]. For vision correction during this kind of pathology, approaches are currently being actively developed in contemporary medicine, aimed at preserving the original photoreceptors and RPE [120] and/or striving to replace cells by activating endogenous regeneration [121] or through cell transplantation [120,122]. One approach to treating a number of degenerative retinal diseases, including age-related macular degeneration, is cell replacement therapy. RPE cells derived from human embryonic SCs and iPSCs are already undergoing clinical trials and have great promise for the treatment of both age-related macular degeneration [123,124,125] and hereditary RPE-associated retinal dystrophies [126]. Despite the highly effective protocols for obtaining RPE cells in sufficient quantities for transplantations that have been developed, they still remain labor- and time-consuming [127]. In addition, there is the problem of immunosuppression and tumorigenesis [128,129], and moral and ethical issues regarding the use of ESCs remain unresolved [130].

Unlike ESCs and iPSCs, adult human RPESCs, despite their limited proliferative potential, do not form tumors [100]. RPESCs are capable of producing RPE-RPESC progeny, which corresponds to the native RPE in its morphological and functional characteristics [95,97,99,113,114]. The preclinical transplantation of RPE derived from RPESCs has shown that an intermediate stage of RPE differentiation is more effective in restoring vision [97]. The successful transplantation of human fRPESCs has been demonstrated on animal models, where the cells differentiate into both RPE and photoreceptors [27].

The heterogeneity of RPE-RPESCs is manifested not only in the functional specialization of cells, but also in the ability of RPE-RPESCs to successfully transplant. Several subpopulations of RPE-RPESCs have been identified that may be potential candidates for effective transplantation [71]. These cells showed the enrichment of signaling pathways associated with cell differentiation and proliferation. Nevertheless, among them, only one cluster of EZH2-positive RPE-RPESCs was isolated, which is capable of integrating into the host RPE monolayer upon transplantation. A clear difference between the transcriptomes of transplant-effective and transplant-ineffective subpopulations of RPE-RPESCs was identified. In addition, molecular pathways associated with graft effectiveness were shown, and potential biomarkers for effective RPE cultures were established [71]. As a primary biomarker of transplantation success, the long noncoding RNA (lncRNA) Three Prime Repair Exonuclease 1 (TREX) was considered, which regulates various cellular processes, including migration and cell survival [71]. These data will allow us to identify from the heterogeneous populations of RPESC-RPE, iPSC-RPE, ESC-RPE, and RPE obtained from other possible cell sources a separate subpopulation of cells that is most suitable for successful transplantation.

Currently, RPE derived from RPESCs represents a potentially unlimited source of HLA-compatible cells and an unlimited donor source with several qualities favorable for transplantation, including stability, ubiquity, and cost. In addition, active phase 1 clinical trials are being carried out in patients with dry AMD (NCT04627428) [131]. It is important to note that transplantation can cause a number of complications, which can be triggered by surgical damage to the retina, leading to its detachment [132]. In this regard, the stimulation of endogenous adult RPESCs present in the eye to produce a new autologous RPE in situ without surgery may be promising for the treatment of many degenerative-dystrophic diseases of the human retina and will help to avoid the above limitations and complications.

## 7. Discussion

The RPE cells in the eyes of vertebrates perform important functions in maintaining the functioning of the neural retina. Damage to RPE cells often causes dysfunction of photoreceptors and neurodegeneration of retinal neurons, which in mammals and humans leads to a weakening and even loss of the vision function [5,133].

Morphometric studies carried out over several decades have shown that the monolayer of RPE cells is not homogeneous [1,28,29,31,40,41]. Recent studies have made it possible to see the heterogeneity of RPE cells, the manifestation of which largely determines the interaction of the RPE with photoreceptors [1,29,45]. In areas where there is a high density of photoreceptor cells in the retina, for example, in the macular region of the retina, the RPE consists of small, tightly adjacent, regular hexagonal cells. In the area where the density of the photoreceptors decreases, the RPE cell density also decreases, while the integrity of the monolayer is maintained due to an increase in the cell area [32,35,36]. The mosaic of photoreceptors also affects other morphological characteristics of RPE cells: multinucleation, pigmentation, the structure of the apical surface, and the shape of the villi. Rod photoreceptors prefer multinucleated RPE cells, while cones prefer mononuclear ones [29,39].

The diversity of morphological cell subpopulations observed in the RPE may vary between individuals. Naturally, this heterogeneity of cells is a reflection of their functional characteristics. Not only the mosaic of photoreceptors, but also their functional load in the retina leaves an imprint on RPE cells. In the macular region, we can observe a high density of photoreceptors and a more intense phototransduction than in the peripheral areas of the retina, and, therefore, the RPE cells in the macula are adapted to an enhanced digestion of the outer segments and to photo-oxidation processes [17,22,44,48,58].

Regarding the genes that control the phenotypic characteristics of cells, it is known that they are tightly regulated and have a stable expression, so even small changes in their expression can produce biologically important differences [25]. Concurrently, gene expression atlases of human RPE cells show similarities between the overall gene expression profiles of the macular and peripheral RPE. A number of researchers believe that the key factor determining function is the genotype, rather than the positional information associated with the topographic position of a cell [2,20,21,25]. Other studies show different levels of gene expression and protein synthesis within the RPE cell layer, depending on the topographic location of the cells themselves and their subpopulations (Table 2). In addition, there is a marked homeostatic heterogeneity of RPE cells, in terms of energy generation, motility, RNA processing, and catalytic processes [26]. All these data indicate that, despite the common functions of human and mammalian RPE cells, individual specialization is present in the selected cell subpopulations (Table 3).

These subtle functional differences are hypothesized to be the result of cellular mosaicism: a phenomenon in which the tissue is a “patchwork” of cells that differ in phenotype or genotype (or both). Moreover, in each “patch”, the neighboring cells are identical, but the neighboring “patches” are different [31]. The functional heterogeneity of the RPE is hereditary in nature, embedded in the genetic program of the cells themselves [71], and dictated by the conditions of the cellular microenvironment [31]. The mosaicism of the RPE is apparently largely determined by epigenetic processes regulating the activation or silencing of genes that accompany normal RPE development and postnatal cell aging [31]. It is assumed that there is a connection between the increasing complexity of the functions of RPE subpopulations and the differentiation duration of the constituent cells [2]. RPE differentiation occurs unevenly, in the radial direction from the center to the periphery and within the layer, since it correlates with the development of different types of photoreceptors. As a result, those RPE functions that are associated with providing photoreceptor functions are dictated by the specific needs of rods and cones: all of this probably underlies the heterogeneity of RPE cells. Therefore, the formation of functionally different RPE subpopulations, which are identified using scRNA-Seq, may be determined by the topographic distribution of photoreceptors and their mosaic. This is reflected in the topographical features of the maintenance of RPE cells and the functioning of photoreceptors, as well as in the response of RPE cells to the influence of various factors [2,53]. Thus, it was shown that RPE cells at different stages of aging demonstrate a different susceptibility to monogenic and polygenic retinal diseases and to the effects of various agents, depending on their topographic location in the layer and their microenvironment. RPE cells react differently to pathological changes during certain chronological periods of ontogenesis [1,25,26].

In the general population of adult human eye RPE cells in vitro, the expression of stem cell pluripotency genes NANOG, OCT4, SOX2, SSEA-4, KLF4, C-MYC, and LIN-28 was detected [100,105,106,108,110]. In another study, OCT4 was not detected in the RPE of the adult human eye [100]. On the one hand, differences in the detection of OCT4 cells in the RPE may be due to the fact that these cells belong to a population of extremely rare primitive oct4 stem cells that are found in the mouse brain [134]. On the other hand, the different results may be explained by zonal differences, and, accordingly, by the contribution of the subpopulations of the RPE cells analyzed in in vitro systems. It cannot be excluded that oct4 can be expressed in pulses, as has been shown in the examples of Müller cell transdifferentiation in the retinas of mice and zebrafish [135,136]. However, OCT4 expression may meet a stoichiometric requirement that will allow cells to enter a multipotent state [137,138]. SOX2 and KLF4 are members of a small group of transcription factors that can restore differentiated cells to a pluripotent state and are major regulators of neural progenitor cells [137,138,139]. The expression of pluripotency genes in RPE cells may indicate the presence of stem cells in this tissue and the possibility of reprogramming the RPE after isolation.

The data on the possibility of the targeted reprogramming of RPE cells are confirmed in RNA-Seq and scRNA-Seq studies of the functional heterogeneity of the native RPE cells of mice and humans, in which subpopulations of cells expressing stem/progenitor cell genes were found [2,53,67]. The number of RPESCs was determined in in vitro experiments, from which it became clear that from 0.003 to 2% of cells are capable of proliferation, self-renewal, and the expression of specific genes characterizing stem cells [26,27,67,100]. Depending on microenvironmental conditions, RPESCs can remain quiescent in a stemness state or exhibit multipotent differentiation under the influence of specific factors. Under expansion culture conditions, when RPESCs self-renew, they generate stem cell progeny and, conversely, under differentiation culture conditions, they generate progeny of various cell types [95,97,99,113,114]. RPESCs can produce RPE cells and, under specific conditions, are capable of generating different types of photoreceptors and nerve cells or mesenchymal cells [27,86,96,100,108].

It is obvious that RPE stem cells are not primitive cells but are differentiated cells, morphologically indistinguishable from the neighboring cells. The stem state of RPESCs maintains a well-structured niche microenvironment consisting of Bruch’s basement membrane, intercellular lateral contacts in the RPE, and apical contacts with the outer segments of photoreceptor cells. Cell proliferation is controlled by the niche components in interaction with endogenous signals [83,86,140]. Like neural stem cells in the brain [141], RPE stem cells are quiescent and require changes in the microenvironment and signals in the niche for their activation and proliferation. The factors that activate the reprogramming of RPE cells and dormant stem cells are the same. This means the loss of intercellular contacts and contacts with surrounding tissues, the remodeling of the cell surface and cytoskeleton, the displacement of RPE cells from their stabilizing environment (niche), proliferation, migration, genome reprogramming, and, ultimately, the sequential acquisition of a different phenotype [140].

The RPE maintains tissue homeostasis through endogenous regulatory systems [6]. Mature RPE can be compared with classical barrier epithelia, which are characterized by a constant cellular turnover throughout life, such as the corneal epithelium, skin epidermis, or intestinal epithelium [83]. To ensure the maintenance of tissue homeostasis, the rate of cell removal must match the production of new cells, which is achieved primarily through stem and progenitor cells. Mature RPE maintains tissue homeostasis mainly through long-term cell survival, although cell density decreases with age, especially in the peripheral RPE. The decrease in cell density is thought to be due to the continued growth of the eye during the juvenile period, and the gradual loss of senescent cells [40,43,46]. Maintaining the integrity of the RPE is achieved through hypertrophy without loss of contact with neighboring cells and/or through cellular replacement (migration from the periphery), which may explain the relative preservation of the middle part of the RPE, and a sharp decrease in the cell density in the periphery [32,142]. The long life of RPE cells does not exclude the possibility of the presence of a few stem/progenitor cells among them, although the question of their localization remains open. It is assumed that less differentiated RPE cells retain the ability to transdifferentiate into other cell types [83]. RPESCs should be located in a topographic zone with less differentiated cells, where there are less rigid cell contacts; this is the peripheral RPE [57], which is as close as possible to the ciliary marginal zone. From an evolutionary point of view, it would seem that RPESCs can be preserved in the ciliary marginal zone of the eye [88]. It was also suggested that the RPE subpopulation of the concentric zone P4 may contain a pool of RPE cells that retain the ability to proliferate [1]. In contrast to these assumptions, the scRNA-Seq results clearly showed that less differentiated cells are found in the central area of the RPE, in particular, in the macular region [2,67]. In the cells of this area, a high expression of stemness and stem cell maintenance genes was found, against the background of a high expression of melanogenesis genes, which indicates an ongoing process of cell development [67]. It can be hoped that experiments with RPE cells from the topographic zones identified to date [1], in the macular (P1) and peripheral (P4 and P5) regions, will help to answer the question about the localization of RPESCs more definitively.

Stem cells are a heterogeneous population, including many variant phenotypes, which complicates these studies [87,141,143]. Therefore, it is not yet possible to prove the similarity of or distinguish between cells capable of transdifferentiation and stem cells in the RPE, and this remains to be understood in the future. It is now clear that mammalian and human RPE stem cells are nevertheless present, and they are a very valuable and interesting research material for a cell source in replacement therapy for the treatment of a number of neurodegenerative diseases of the human eye. Dr Sally Temple spoke about this in her talk at the annual meeting of the Association for Research in Vision and Ophthalmology (ARVO) at the National Eye Institute (NEI) in May 2022: “In 2012 we described an adult RPE stem cell (the RPESC) present throughout life, even in patients in their 90s. We demonstrated that adult RPESCs can be dramatically expanded in vitro to create cells with key physiological characteristics of the native RPE layer and sufficient cells for hundreds of doses. Importantly, they can be transplanted successfully as a mature RPE monolayer in animal models. In addition, they can be transplanted as a suspension of intermediate progenitor cells that are post-mitotic but not fully differentiated; after subretinal injection into the RCS rat of retinal degeneration, adult RPESC-derived RPE progenitor cells prevented photoreceptor and vision loss. These adult RPESC-derived cells could successfully integrate into the existing RPE layer and persist long-term in animal models without adverse safety findings, supporting their use as RPE cell replacement therapy. Adult RPESC-RPE cells in suspension are currently in clinical trial as allogeneic therapy for patients with dry AMD” [131]. It is hoped that further research will fully reveal the potential of RPESCs and the possibilities of their medical use.

## 8. Conclusions

To summarize, it should be noted that, behind its apparent simplicity, the monolayer of RPE cells that is vital for the retina and that can be metaphorically compared to a cobblestone pavement hides an amazing diversity of cells, from terminally differentiated cells to highly plastic cells and stem cells. The complex structure of the eye and the specific organization of the visual process determine the morphological differences between RPE cells. At the same time, the types of photoreceptors and the mosaic of their distribution also dictate differences in RPE functions. A major breakthrough in the study of RPE functions is provided by the RNA-Seq and scRNA-Seq methods, which make it possible to reveal the specificity of gene expression in RPE cells. At the same time, one must be aware that the morphofunctional and molecular genetic features of these heterogeneous cells have yet to be studied in detail. These studies will provide a deeper understanding of the role of RPE cells both in the norm and during the development of pathological processes in the retina, and will undoubtedly be important for regenerative ophthalmology.

## Figures and Tables

**Figure 1 cells-13-00281-f001:**
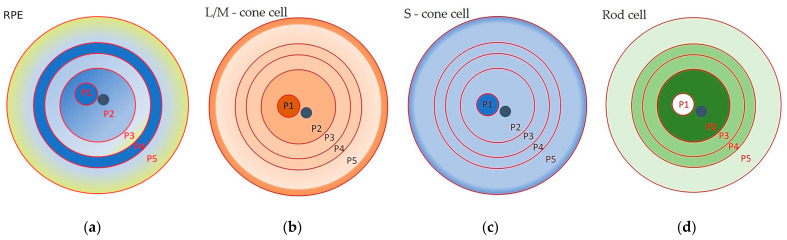
Schematic representation of a heat map of the retinal pigment epithelium (RPE), and a diagram of the distribution of photoreceptors of the human retina: (**a**) concentric zones of the adult human eye RPE: P1—fovea and parafovea; P2—perifovea; P3—the immediate periphery (or equatorial region); P4—a ring of RPE cells identified for the first time, resembling the RPE cells of zone P2; and P5—the far periphery, ending with the ora serrata; the dark blue staining indicates small hexagonal RPE cells of regular shape; a decrease in the blue color saturation with a transition to yellow indicates that the cell area of the RPE increases, cell hexagonality is disrupted, cells become more elongated, and cell density decreases in the monolayer; (**b**) the concentric zones of M and L cone distribution in the adult retina; (**c**) the concentric zones of S cone distribution in the adult retina; and (**d**) the concentric zones of rod distribution in the adult retina. Interpretation has been conducted in accordance with [1] and [38]. The more saturated the color, the higher the cell density. RPE—retinal pigment epithelium; L/M—Long wavelength/Medium wavelength cones; and S—Short wavelength cones.

**Table 1 cells-13-00281-t001:** Morphometric analysis of the adult human retinal pigment epithelium (RPE) monolayer based on data from [1].

Topographic Zone	Concentric Zones	Distance from the Optic Nerve, mm	Average Area, µm^2^	Aspect Ratio	Hexagonality	Number of Neighboring Cells	Rod/Cone to RPE Ratio
**Central** **(macular) RPE**	P1	a 3 mm wide spot	147.24 ± 15.36	1.15 ± 0.04	9.31 ± 0.11	5.55 ± 0.35	0.002
	P2	0–10	201.74 ± 17.45	1.18 ± 0.02	9.25 ± 0.05	5.47 ± 0.43	0.129 (0.068 for the perifovea)
**Peripheral RPE**	P3	10–14	231.21 ± 18.08	1.23 ± 0.03	9.12 ± 0.08	5.46 ± 0,63	0.087
	P4	14–17	176.76 ± 18.68	1.27 ± 0.04	9.00 ± 0.12	5.64 ± 0.25	0.096
	P5	17–24	331.87 ± 27.23	1.33 ± 0.03	8.79 ± 0.11	5.04 ± 0.46	0.026

## Data Availability

Not applicable.

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
