# Peer review of "Recent Achievements in the Heterogeneity of Mammalian and Human Retinal Pigment Epithelium: In Search of a Stem Cell"

_cells, 2024, doi:10.3390/cells13030281_

Round 1
Reviewer 1 Report
Comments and Suggestions for Authors
The authors present a review concerning the morphological, topological and gene expression differences in the RPE populations of the human (which has a fovea/macular region) and non-primate mammalian species. They incorporate sc-seq and other formats to try to present a comprehensive review of the associations between morphology, function and topology. This is an important review subject, and the authors are to be commended for attempting it. However, the report is more of a 'book report' format, which needs to be extensively revised and re-done before it is resubmitted. the writing is extremely unclear and makes it hard to follow for any reader. The reasons are these:
1.The report is extremely difficult to follow, with little regard for condensation of important facts (necessary in a comprehensive review) and a consistent flow of different facets (morphology, topology, etc). This makes it both tiring and the normal reader is likely to become confused as to what important fact(s) the authors are trying to make. They are not likely to follow every section, and the repetitive nature of the facts presented in the different sections will make it hard to distinguish what points they are trying to make.
2. The format of the report is repetitive, and the same information appears numerous times in different sections, without trying to differentiate why this is appearing.
3. The authors need to present the sections in ways that minimize the use of the same information and genes in multiple sections (eg: why use NANOG, OCT4, SOX2, etc in so many sections? and what would the expression of these common genes (which are stem-cell associated genes) mean, in terms of the different regions or functions, rather than simply being expressed in many areas.
4. The manuscript suffers from both tables that are poorly organized (tables 1-3) and too large in terms of simple listing. They could be condensed and better organized, to make a point. Table 2 simply lists the specific genes expressed by human RPE and the function of these genes in the different regions. Just giving the name is sufficient; the reader will be able to work out the function. It would also be best to set up the table to compare the relative similarity/expression of human and mouse, in the different regions, to compare relative functions.
It is the job of the authors to simplify a complex problem and make the article accessible to readers. Condensing the manuscript, focusing the sections to specific functions, and trimming the repetitive nature would make it more accessible to interested readers.
Other points:
1. The numerous uses of 'thus', 'so's, and 'the data allowed for' , etc, make it very tiring to read, as does the use of both the name (Choi et al,) and the reference number. This approach should be revised to simply state the fact, and give the reference number.
2. Delete lines 110-115. This is too much data for a general review.
3. Section three has a tremendous amount of detail (lines 155-174). This should be summarized/condensed to simple description or reference.
4. Lines 185-187. What has this to do with anything?
5. Mover section 4.1 to the front. In particular second paragraph of 4.1 should be moved to the front after the initial paragraph of the introduction.
6. What is a 'chronological development sequence'?
7. Greatly condense table 2.
8. Lines 339-351: put mouse info into table 1 for comparison with human and cite in lines 352-362.
9. Section 5/ line 385: What do the authors mean by this statement? That small changes in early cell types lead to permanent changes and deviant cell lines? This is an important question re: Natural vs IPSC derived RPE. Please make this clearer: What is the key problem with IPSC's derivation?
Important findings (416-424) are hidden by a lot of irrelevant discussion in 424-430.
10. SEction 6: Delete 436-441
11. In line 456: pleae define what is meant by plastic changes. Is this stem-ness?
12. Line 473-475. I think neuronal transdifferentiation only occurs in amphibians that examples that the authors give are all amphibian. But the paragraph leads one to believe that mammals can exhibit this too. If there is a true mammalian example, then give it. I would rewrite the entire sevction to strengthen the arguement.
line 546: Valproic, not valpolic.
13. 567-578 could be put in a comparison table. And what is the significance of lines 590-599? This section should be condensed significantly.
14. Lines 627-630 seem to be a duplication of lines 530-533
15. What are the important facts of lines 658-665? That no bipolar or horizontal markers were found? And what do lines 662-665 mean?
16. An explanatory figure for 6.6 would be nice. And are the authors trying to give the importance of this section to use in retinal regenerative surgery? If so, they should say when this would be done, the problems associated with using one vs another type, or optimal time of development.
17. section 6.7. This is an important section, but in Lines 698-713: simplify and give the meaning.
18. 774-777: Where is the reference?
19. 797-800: This appears repetitive with lines 779-785.
20. Lines 837-852: What is the reason for repeating this previously stated material again in the text?
Comments on the Quality of English Language
Repetitive sections, citing of author names and then reference numbers.
Author Response
Dear Reviewer,
We thank you very much for taking the time to review this manuscript. We appreciate your insightful and detailed analysis of our review. In general, we agree with your comments and recommendations. We have revised our manuscript according to your comments. Please find our responses below, in the uploaded revised file and the corresponding revisions/corrections highlighted/in track changes in the re-submitted version of our manuscript.
Thank you for your review and questions.
Sincerely,
Authors of the manuscript

Reviewer 2 Report
Comments and Suggestions for Authors
It is a well written review of heterogeneity of mammalian and human RPE. However, my question is reproducibility of these studies. If we see such considerable plasticity of the cells, how general could be identifications of zones, and how well zones will be reproduced in different experiments. What are the differences and similarities of results from different labs using the same species and cell cultures? I would like to see some statistics analysis of reproducibility.
Comments on the Quality of English Languagefine
Author Response
Dear Reviewer,
We thank you very much for taking the time to review this manuscript. We thank your insightful and detailed analysis and positive assessment of our review.
We have revised our manuscript according to your comments. Please find our responses below, in the uploaded revised file and the corresponding revisions/corrections highlighted/in track changes in the re-submitted version of our manuscript.
Sincerely,
Authors of the manuscript

Reviewer 3 Report
Comments and Suggestions for Authors
The review by Rzhanova et al is an interesting overview of the heterogeneity of retinal pigment epithelium in mammals. The authors reported in detail the molecular, genetic, and morphological characteristics of this heterogecneity, emphasizing the possibility of its use in regenerative medicine.
I advise the authors to improve the readability of Table 1 which is a bit confusing, otherwise the paper can be accepted without reservation.
Author Response
Dear Reviewer,
We thank you very much for taking the time to review this manuscript, your detailed analysis of our article and your comments. According to your comments and recommendation, we have removed the unnecessary data from Table 1, and the unnecessary explanations which are presented in the text before the table. The table caption has been also modified and shortened.
Sincerely,
Authors of the article

Reviewer 4 Report
Comments and Suggestions for Authors
In this manuscript, Rzhanova et al provide a comprehensive review of the recent discovery of RPE heterogeneity in various mammalian species. The review delves into the morphometric heterogeneity observed in human RPE cells (Ortolan et al., 2022) and the molecular heterogeneity identified in both human and mouse RPEs by the research groups of Xu et al. (2021), Lee et al. (2022), and Pandey et al. (2022). The authors also discuss the presence of stem/progenitor cell-like properties, such as cell-renewal/proliferation and pluripotency, within cultured RPE cell population. Additionally, they provide a comprehensive review of three modes of stem cell behavior that facilitate the plasticity of differentiation. Finally, the authors discuss how these studies could contribute to the understanding/development of RPE stem cells and their potential therapeutic value in the field of regenerative ophthalmology. This review paper is an excellent and timely summary for RPE biology and stem cells.
Author Response
Dear Reviewer,
We thank you very much for taking the time to review this manuscript and for your detailed analysis and appreciating of our manuscript.
Sincerely,
Authors of the article

Round 2
Reviewer 1 Report
Comments and Suggestions for Authors
The authors have eliminated many of the redundant paragraphs, but have done nothing to actually revise the use of both authors names and references, and have left the manuscript with dangling sentences (see line 290). This needs to be cleaned up extensively, and I would appreciate a PDF copy with the revisions, but without the strikethroughs, so that I can appreciate the English sentence structure. Otherwise, I do not accept this manuscript in its current form.
Comments on the Quality of English LanguageThe authors have eliminated many of the redundant paragraphs, but have done nothing to actually revise the use of both authors names and references, and have left the manuscript with dangling sentences (see line 290). This needs to be cleaned up extensively, and I would appreciate a PDF copy with the revisions, but without the strikethroughs, so that I can appreciate the English sentence structure.
Author Response
Response to Reviewer 1
Dear Reviewer,
We thank you very much for taking the time to review and detailed analysis of our manuscript. We agree with your comments and recommendations. Please find the uploaded revised file in PDF copy with the corresponding changes in the re-submitted version of our manuscript.
Those parts of the text that have been edited were marked in red font.
- We have revised the use of the authors names in the text and left only the corresponding references.
- We have eliminated the dangling sentences (line 290) in the text to save arrange the lines in the appropriate sequence.
- We have made corrections to the English sentences structure and also eliminated errors and typos.
Thank you again for your review, comments and questions. We hope that we have taken into account all the comments and edited the text in accordance with them.
Please see the attachment.
Sincerely,
Authors of the manuscript
23.01.2024
